# Modeling the response of Northwest Greenland to enhanced ocean thermal forcing and subglacial discharge

Mathieu Morlighem[1], Michael Wood[1], Hélène Seroussi[2], Youngmin Choi[1], and Eric Rignot[1,2]

[1]University of California, Irvine, Department of Earth System Science, 3218 Croul Hall, Irvine, CA 92697-3100, USA
[2]Jet Propulsion Laboratory - California Institute of Technology, 4800 Oak Grove Drive, Pasadena, CA 91109-8099, USA

**Correspondence:** Mathieu Morlighem (mathieu.morlighem@uci.edu)

**Abstract.** Calving front dynamics is an important control on Greenland's ice mass balance. Ice front retreat of marine-terminating glaciers may, for example, lead to a loss in resistive stress, which ultimately results in glacier acceleration and thinning. Over the past decade, it has been suggested that such retreats may be triggered by warm and salty Atlantic water, which is typically found at a depth below 200-300 m. An increase in subglacial water discharge at glacier ice fronts due to enhanced surface runoff may also be responsible for an intensification of undercutting and calving. An increase in ocean thermal forcing or subglacial discharge therefore has the potential to destabilize marine terminating glaciers along the coast of Greenland. It remains unclear which glaciers are currently stable but may retreat in the future, and how far inland and how fast they will retreat. Here, we quantify the sensitivity and vulnerability of marine-terminating glaciers along the northwest coast of Greenland (from 72.5° to 76°N) to the ocean forcing and subglacial discharge using the Ice Sheet System Model (ISSM). We rely on a parameterization of undercutting based on ocean thermal forcing and subglacial discharge, and use ocean temperature and salinity from high-resolution ECCO2 (Estimating the Circulation & Climate of the Ocean, Phase II) simulations at the fjords mouth to constrain the ocean thermal forcing. The ice flow model includes a calving law based on a tensile Von Mises criterion. We find that some glaciers, such as Dietrichson Gletscher or Alison Gletscher, are sensitive to small increases in ocean thermal forcing, while others, such as Illullip Sermia or Cornell Gletscher, are remarkably stable, even in a 3-degree ocean warming scenario. Under the most intense experiment, we find that Hayes Gletscher retreats by more than 50 km inland by 2100, into a deep trough, and its velocity increases by a factor of 3 over only 23 years. The model confirms that ice-ocean interactions can trigger extensive and rapid glacier retreat, but the bed controls the rate and magnitude of the retreat. Under current oceanic and atmospheric condition, we find that this sector of the Greenland ice sheet alone will contribute more than 1 cm to sea level, and up to 3 cm by 2100 under the most extreme scenario.

# 1   Introduction

Over the past two decades, many glaciers along the northwest coast of Greenland have been retreating and accelerating, sometimes dramatically (e.g., Moon et al., 2012; Wood et al., 2018). It has been suggested that the retreat of these glaciers is initiated by the presence of warm, salty, subsurface Atlantic Water (AW) in the fjords (e.g., Straneo et al., 2010; Straneo and Heimbach, 2013; Rignot et al., 2012; Holland et al., 2008). This water is typically found 200 to 300 m below the surface (e.g., Rignot et al., 2016a; Holland et al., 2008). Surface runoff has also been increasing over the past decades (van den Broeke et al., 2009; Fettweis et al., 2013b; Tedesco et al., 2013), which enhances subglacial water discharge at the base of calving fronts. This freshwater flux enhances the circulation of the ocean in the fjord (Xu et al., 2012), which in turn further increases the melting rate, and therefore the rate of undercutting at the calving face of marine terminating glaciers. While we expect both surface runoff and the ocean heat content to continue to increase over the next century, it remains unclear how they are going to affect ice dynamics and the ice discharge into the ocean.

While geographically close, individual outlet glaciers along the coast respond differently to frontal forcing. It has been proposed (e.g., Wood et al., 2018; Catania et al., 2018) that this heterogeneity in glacier behavior may be due to differences in bed topography and fjord bathymetry, which may prevent the access of AW to interact with calving fronts due to the presence of sills in the fjord. It has also been suggested that many glaciers are currently resting on pronounced ridges, or in regions of lateral constrictions, which stabilizes the glaciers' calving fronts, and prevents warm water from dislodging them from their current position (Catania et al., 2018). The idea that ice front dynamics is, to a large extent, controlled by subglacial topography has first been investigated in Alaska (Mercer, 1961; Meier and Post, 1987) and has more recently been extended to Greenland (e.g., Warren, 1991; Warren and Glasser, 1992; Carr et al., 2015; Lüthi et al., 2016). It is not certain to which degree the glaciers of the northwest coast remain sensitive to enhanced thermal forcing from the ocean: some glaciers are on the verge of a fast and extensive retreat, others may continue retreating at the same rate, and some may remain stable. Numerical modeling can help us assess the sensitivity of these individual glaciers to ocean temperature along the coast, and their potential for fast retreat and mass loss, affecting sea level rise.

While many model-based studies have been focusing on the response of the Greenland ice sheet to climate change, they either did not include moving calving fronts (e.g., Bindschadler et al., 2013; Gillet-Chaulet et al., 2012), or were based on flow-line models (e.g. Nick et al., 2013) that do not capture changes in lateral drag well (since lateral drag is parameterized) or the complex three-dimensional shape of the bed that affects the retreat rate (Choi et al., 2017; Bondzio et al., 2017), and did not consider undercutting. Here, we want to overcome these limitations by using a plan-view model with a moving calving front. The calving front position is allowed to move and is a function of ice velocity, calving rate and rate of undercutting. While much progress has been made in terms of capturing ice flow through improved datasets (Aschwanden et al., 2016) and through the development of new stress balance solvers not based on the Shallow Ice Approximation, calving and undercutting remain areas of active research. We use two existing parameterizations of ocean undercutting (Rignot et al., 2016b) and calving (Morlighem et al., 2016). While these parameterizations are approximations and do not include all the physics involved in ice-ocean interactions, they have been tested with reasonable success on several glaciers of Greenland (e.g. Morlighem et al.,

2016; Choi et al., 2017; Bondzio et al., 2018; Rignot et al., 2016b). The objective of this study is not to make projections, as we are not forcing the model with given Representative Concentration Pathway (RCP) scenarios, but to assess the sensitivity of Northwest Greenland using existing parameterizations for iceberg calving and undercutting.

We focus here on the northwest coast of Greenland between 72.5° and 76°N: from Upernavik Isstrøm to Sverdrup Gletscher (figure 1). This is one of the regions of Greenland where the bed is remarkably well constrained by ice thickness measurements from NASA's Operation IceBridge mission (Morlighem et al., 2017), and where NASA's Oceans Melting Greenland mission has been collecting multibeam bathymetry data in most fjords.

We first describe the numerical model and then run the model to 2100 under different scenarios of increase in ocean thermal forcing and subglacial discharge. We then discuss the implications of these experiments, the model limitations, and make recommendations for future model studies.

## 2  Method and data

### 2.1  Ice flow model setup

We use the Ice Sheet System Model (ISSM, Larour et al., 2012) and initialize the model with conditions similar to 2007, which is the nominal year of the surface digital elevation map used here (gimpdem, Howat et al., 2014). The ice surface elevation and bed topography are from BedMachine v3 (Morlighem et al., 2017), and we use satellite derived surface velocities from Joughin et al. (2010) to invert for basal friction, following Morlighem et al. (2010). We use a Shelfy-Stream Approximation (SSA, MacAyeal, 1989) for the ice stress balance. While not accurate in slow moving regions, this model is an excellent approximation for the fast outlet glaciers (i.e. $> 200$ m/yr) that we are focusing on here, where sliding velocities are significantly larger than deformational velocities (e.g., Rignot and Mouginot, 2012). We assume a depth-averaged viscosity equivalent to a temperature of -8°C, which is consistent with Seroussi et al. (2013), and we use a linear viscous basal friction law following Budd et al. (1979):

$$\boldsymbol{\tau}_b = -C^2 N \mathbf{v}_b, \tag{1}$$

where $\boldsymbol{\tau}_b$ is the basal friction, $\mathbf{v}_b$ is the ice basal velocity, $C$ is a friction coefficient that is inverted for using surface velocities, and $N$ is the effective pressure. For simplicity, we assume that $N$ is equal to the ice pressure above hydrostatic equilibrium, as if the subglacial hydrological system was forming a sheet connected to the ocean. The model mesh comprises 380,000 elements, and its resolution varies between 100 m near the coast and 1 km inland. The model time step is one week.

In order to capture the dynamic motion of the calving front, we rely on the level set method (Osher and Sethian, 1988; Bondzio et al., 2016), where the velocity at which the calving front moves is defined as:

$$\mathbf{v}_{\text{front}} = \mathbf{v} - \left( c + \dot{M} \right) \mathbf{n}, \tag{2}$$

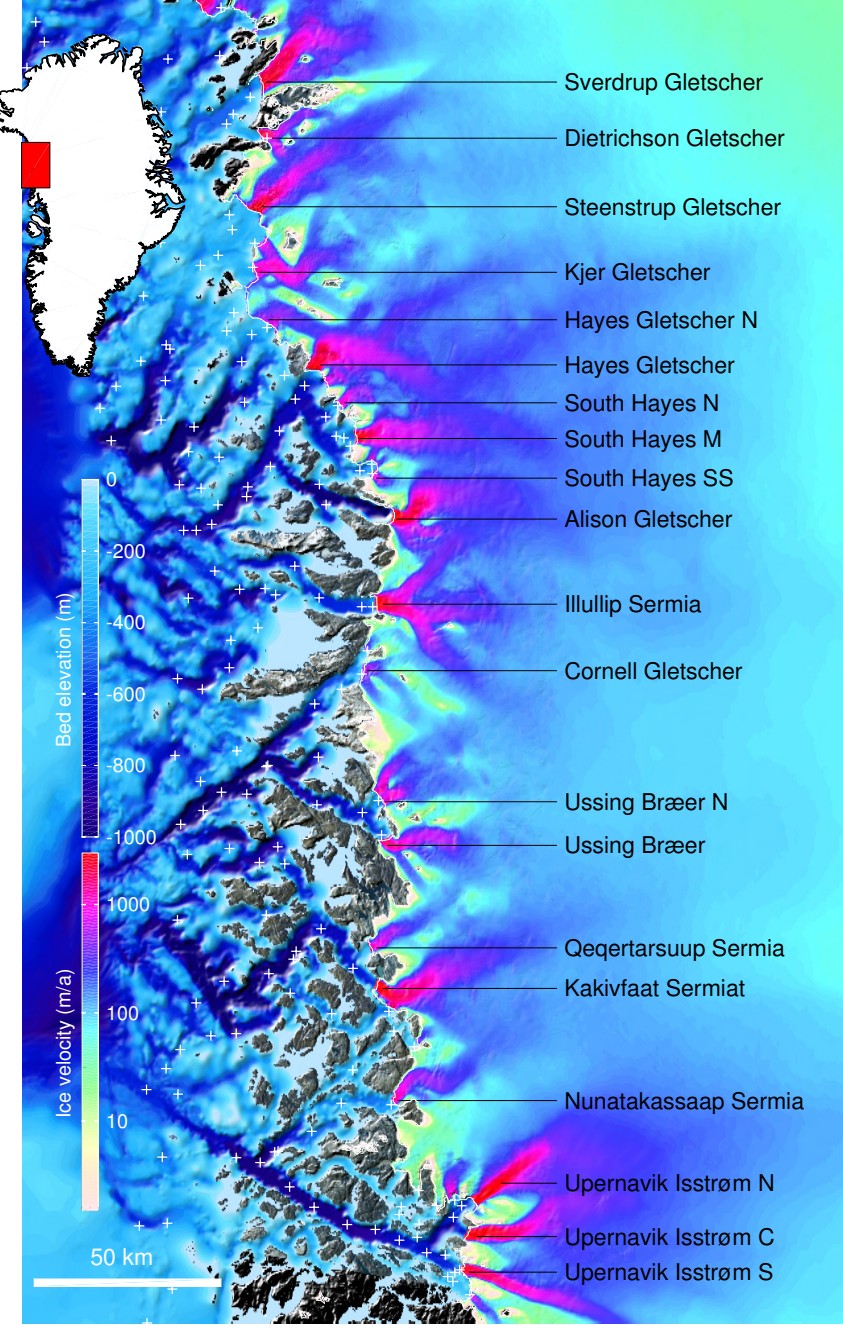

**Figure 1.** Ocean bathymetry (m, blue color scale) and ice velocity (m/a, Joughin et al., 2010) of Northwest Greenland. The white line shows the 2007 ice sheet extent, and white crosses indicate the locations of CTD data from NASA's Oceans Melting Greenland campaign that are used to calibrate the thermal forcing.

where $\mathbf{v}$ is the ice horizontal velocity vector, $c$ is the calving rate, $\dot{M}$ is the rate of undercutting at the calving face, and $\mathbf{n}$ is a unit normal vector that points outward from the ice domain. Much research is currently being dedicated to derive parameterizations for $c$ and $\dot{M}$; here we chose to use two recent parameterizations described below.

## 2.2 Undercutting parameterization

We rely on the undercutting parameterization from Rignot et al. (2016b), where the rate of undercutting (in m/day) at the calving face is assumed to follow:

$$\dot{M} = \left( A h q_{sg}^{\alpha} + B \right) \tilde{T}^{\beta}, \tag{3}$$

where $h$ is the water depth at the calving front (in m), $A = 3 \times 10^{-4}$ m$^{-\alpha}$ day $^{\alpha-1}$ °C$^{-\beta}$, $\alpha = 0.39$, $B = 0.15$ m day$^{-1}$ °C$^{-\beta}$, and $\beta = 1.18$. $\tilde{T}$ is the ocean thermal forcing (in °C), defined as the difference in temperature between the potential temperature of the ocean and the depth dependent freezing point of sea water:

$$\tilde{T} = T - T_F \tag{4}$$

where $T$ is the ocean temperature at a given depth, and $T_F$ is the temperature of the local freezing point, which is assumed to be a linear function of salinity and pressure, following equation (1) of Xu et al. (2012). $q_{sg}$ is the subglacial discharge at the glacier terminus (Rignot et al., 2016b) (in m/day). Both $\tilde{T}$ and $q_{sg}$ are monthly averaged. The coefficients $\alpha$ and $\beta$ are close to the ones expected from the plume theory (Jenkins et al., 2010; Jenkins, 2011), but were determined from a high-resolution ocean modeling study. The introduction of $B$ is necessary to account for the presence of melt in the case where there is no subglacial discharge. The dependence on $h$ was determined from model experiments with different depths and seems to reflect an acceleration of the melt plume when it rises from greater depths (Rignot et al., 2016b).

To estimate the subglacial discharge of melt water, $q_{sg}$, we use the results from the downscaled 1 km RACMO runoff field (Noël et al., 2016) with the subglacial melt rates from Seroussi et al. (2013), and assume for simplicity that the discharge is uniformly distributed across the calving face. Xu et al. (2013) showed that the assumption of uniformly distributed melt generates only a 15% difference in melt compared to a distributed source of $q_{sg}$.

The ocean thermal forcing, $\tilde{T}$, is derived from the Estimating the Circulation and Climate of the Ocean, Phase 2 (ECCO2, 2007-2011) and Phase 4 (2007-2015), following the procedure described in Wood et al. (2018). To account for the presence of sills in the fjord, $\tilde{T}$ is depth averaged between the sea level, and the deepest point for which there is a direct horizontal connection to the fjord mouth. The calculated effective depth assumes an ocean perfectly stratified, and decreases as we get closer to the calving front since ocean currents are potentially blocked by the bathymetry. Figure 2 illustrates the effective depth for the case of Sverdrup Gletscher. Note that we define the effective depth over the entire model domain, even under currently ice-covered regions. If the modeled ice front retreats past a high bump, it will be accounted for in the calculation of the thermal forcing and the rate of undercutting will be reduced (See figures 3b and 3c). Note that this undercutting parameterization facilitates the definition of the rate of undercutting everywhere in the model domain, and its magnitude depends on the ice front location. The ice sheet model is forced by the surface mass balance of RACMO 2.3 averaged between 1961 and 1990:

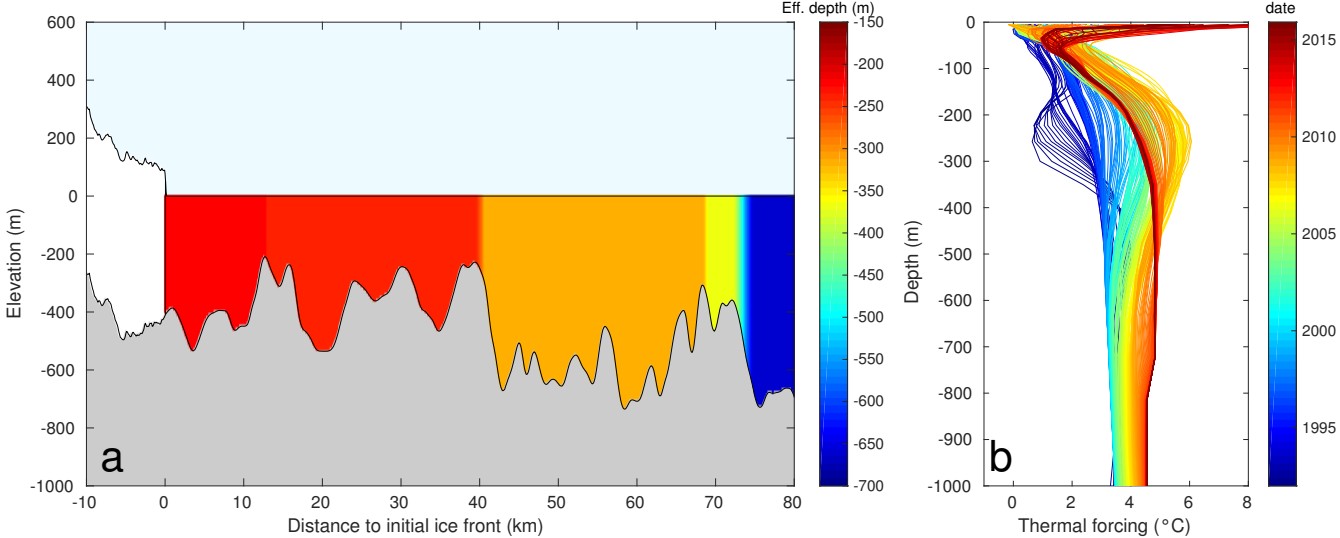

**Figure 2.** (a) Effective depth (m) of the fjord of Sverdrup Gletscher. The effective depth decreases as we go from the fjord mouth ($x = 80$ km) to the glacier terminus ($x = 0$ km). (b) Thermal forcing at the fjord's mouth (°C) for Sverdrup Gletscher from ECCO2.

the increase in runoff (due to the anomaly applied) is assumed to not affect the surface mass balance, but only undercutting through the parameterization provided by equation 3.

### 2.3 Calving parameterization

We assume that the calving rate follows the parameterization proposed by Morlighem et al. (2016), for which the calving rate is proportional to the tensile von Mises stress:

$$c = \|\mathbf{v}\| \frac{\tilde{\sigma}}{\sigma_{\max}} \tag{5}$$

where $\tilde{\sigma}$ is the tensile von Mises stress, as defined in Morlighem et al. (2016), and $\sigma_{\max}$ is a threshold that needs to be calibrated for each basin. This calving law is obviously a simplification that may not capture all modes of calving as it only relies on tensile stresses. It is also assumed here that $c$ and $\dot{M}$ are independent, which is a simplification, but has shown some promising results on real-world applications (e.g., Morlighem et al., 2016; Choi et al., 2017, 2018; Bondzio et al., 2018).

To calibrate the calving threshold, we run the model for 10 years: from 2007 to 2017, using the thermal forcing from ECCO2, and adjust $\sigma_{\max}$ in order to match the extent of Landsat-derived ice front retreat: we try to match the observed retreat from 2007 to 2017 along a central flow line for each glacier, not the retreat rate. This calving threshold is uniform by basin and held constant through time in all simulations. Another possible approach would be to calibrate $\sigma_{\max}$ during a period of ice front stability. One of the problems with this alternative approach is that stable glaciers generally have their termini on distinct basal features, such as ridges or ledges. The numerical model is also stable for a wide range of $\sigma_{max}$ under these conditions, as

shown in Morlighem et al. (2016) and Choi et al. (2018). The threshold $\sigma_{max}$ is easier to calibrate for retreating glaciers, as it directly constrains the rate of retreat.

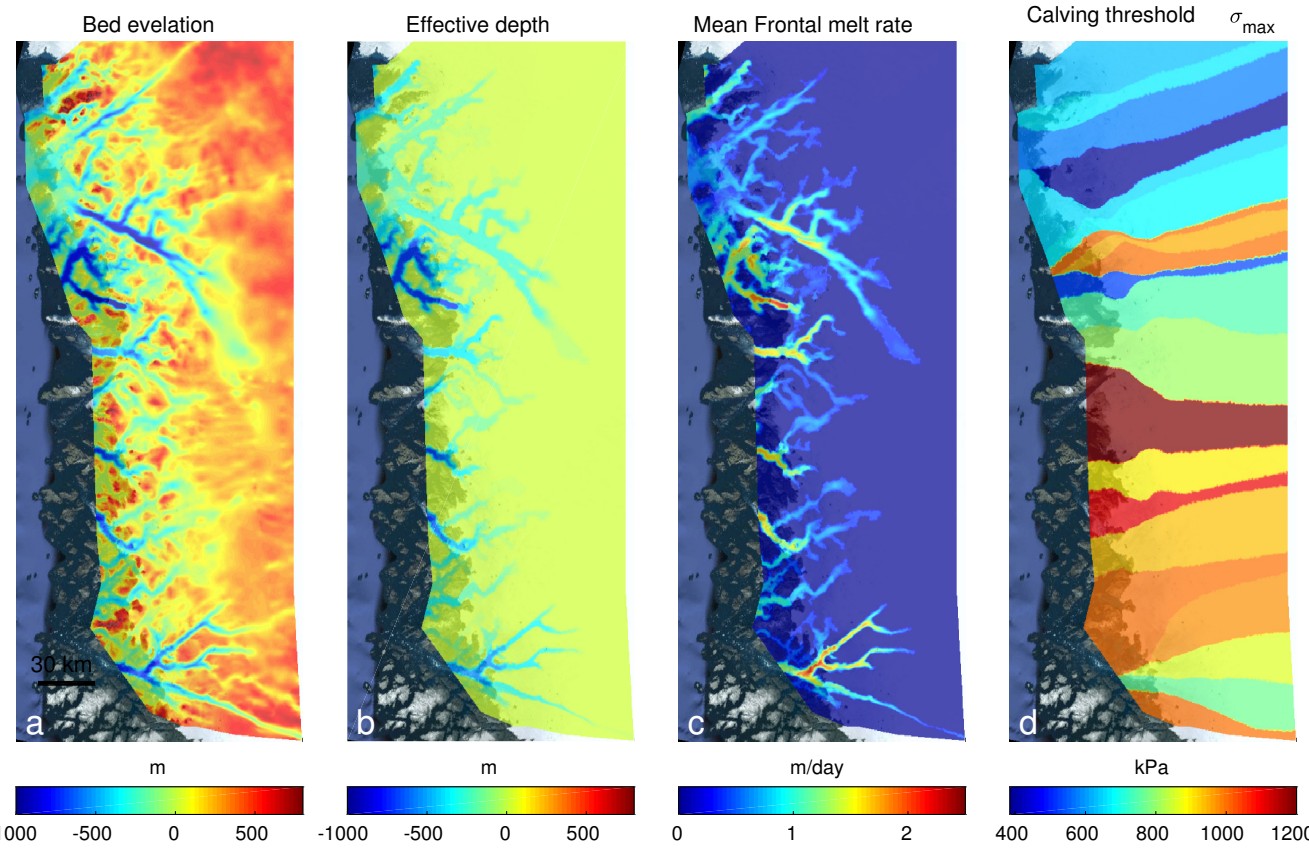

**Figure 3.** (a) bed topography (m), (b) effective depth (m), (c) calculated mean rate of undercutting from 2007 to 2017 (m/day), and (d) calibrated $\sigma_{max}$ (kPa)

## 2.4 Experiments

After this calibration phase, we run the model forward, from 2017 to 2100, under different scenarios of ocean forcings and different scenarios of increase in subglacial discharge. Yin et al. (2011) analyzed the results of 19 climate models to quantify ocean warming around the coast of Greenland over the coming centuries. They found that West Greenland's subsurface ocean temperature reaches between 0.5 and 4°C, with a mean of 1.5°C by 2100. CMIP5 results suggest similar rates of warming by the end of the century under RCP8.5 (D. Slater, pers. comm.). A $+2$°C is also in line with the global atmospheric temperature rise target of the Paris Agreement. Even though there will be a lag in the response of the ocean to atmospheric warming, we do expect that polar amplification could increase ocean temperature further at high latitudes. We therefore consider here a range in $\tilde{T}$ increase from 0 to $+3$°C.

In terms of subglacial discharge, observations over the past decade have shown that surface melting has increased over the entire Greenland ice sheet (van den Broeke et al., 2009; Fettweis et al., 2013b; Tedesco et al., 2013). Fettweis et al. (2013a) showed that meltwater runoff could be multiplied by a factor of 10 by the end of the century. We therefore multiply the subglacial discharge by a factor of up to 10, starting at year 2017.

Overall, we perform here 40 experiments: we increase the ocean thermal forcing, $\tilde{T}$, instantly by increments of 1°C up to 3°C, and multiply the ocean subglacial discharge by up to a factor of 10. We then run the model forward from 2017 to 2100. The rate of undercutting (Equation 3) is therefore modified as follows:

$$\dot{M} = \left(A\,h\,(q_{sg} \times q_a)^\alpha + B\right)\left(\tilde{T} + \tilde{T}_a\right)^\beta, \tag{6}$$

where the subglacial discharge anomaly factor $q_a$ varies from 1 to 10, and the thermal forcing anomaly, $\tilde{T}_a$, varies from 0 to 3°C. From 2007 to 2016, we rely on the thermal forcing ($\tilde{T}$) and subglacial discharge ($q_{sg}$) from ECCO2 and RACMO. For 2017 to 2100, as we do not run a coupled model, we repeat the thermal forcing and subglacial discharge of year 2016 until the end of the century, with the anomalies described above. While a gradual increase in ocean thermal forcing and subglacial discharge would be more realistic, we want here to perform a sensitivity analysis in order to determine the glaciers that are more at risk.

Additionally, we perform a *Control Experiment* where the ice front is kept fixed. This control experiment is designed to quantify the impact of including moving boundaries in future simulations.

## 3 Results

Figure 3d shows the chosen value of the stress threshold over the model domain. For the southern half, we find a stress threshold within 20% of 1 MPa, which is consistent with what was found in other studies (Petrovic, 2003; Morlighem et al., 2016). Over the northern side of the domain, however, the stress threshold has to be decreased to ~650 kPa in order to match the pattern of retreat. This would suggest that the ice is less resistant to tensile stress, but this is more likely an artefact that is due to our underestimation of the rate of undercutting in this region. Wood et al. (2018) noted that the north-south temperature gradient in the ocean model was poorly-represented in this region, and that the resulting thermal forcing was too cold. The model therefore requires a decrease in the stress threshold, thereby increasing the calving rate, $c$, in order to capture the correct amount of ice retreat over the past 10 years. We could have kept $\sigma_{max}$ constant, equal to 1 MPa, and optimize for the ocean thermal forcing instead, but the spatial and temporal variability in $\tilde{T}$ makes its calibration difficult. Optimizing a single scalar parameter per glacier is more practical.

Figure 4 shows ice front positions that were manually digitized from Level 1 Landsat imagery, together with modeled ice front positions between 2007 and 2017 for four glaciers along the coast. The first two columns of table 1 list the observed and modeled retreat for the same time period along a central flow line for the chosen value of the stress threshold. By manually tuning the stress threshold ($\sigma_{max}$) for each basin, we are able to match the retreat of the past 10 years for all 17 glaciers for which a change has been documented, except for Ussing Bræer N (Table 1) for which we model a retreat of almost 3 km instead

of an advance of 300 m. This inconsistency may be due to errors in the bed topography near the front. We note, however, that under all scenarios, this glacier remains remarkably stable at its 3 km retreated position, which coincides with a large bump in the bed topography. Overall, we find that with a unique scalar parameter constant in time for each glacier, the modeled ice front retreat is in very good agreement with observations, which is consistent with Choi et al. (2018). The retreat rate of Dietrichson

5    Gletscher is well captured (figure 4a vs 4b). While the model overestimates the retreat on the southern side of the fjord, there is nonetheless an overall good agreement between the modeled and observed retreat between 2007 and 2017. The front of Illullip Sermia is remarkably stable in both observations and in the model (figure 4c and 4d), as it is currently located on a pronounced sill in the bed topography. The modeled ice front of Upernavik Isstrøm retreats more in the southern half of the fjord than the northern half compared to the observations, but the increase in ice retreat over the past 2 years is captured (figure 4e and 4f).

10   The complex pattern of ice front retreat of Kakivfaat Sermiat is also reproduced with a slight difference in timing (figure 4g and 4h). The 2017 modeled front position is also more retreated than what has been observed, but we find the same strong control of the bed topography in the pattern of retreat.

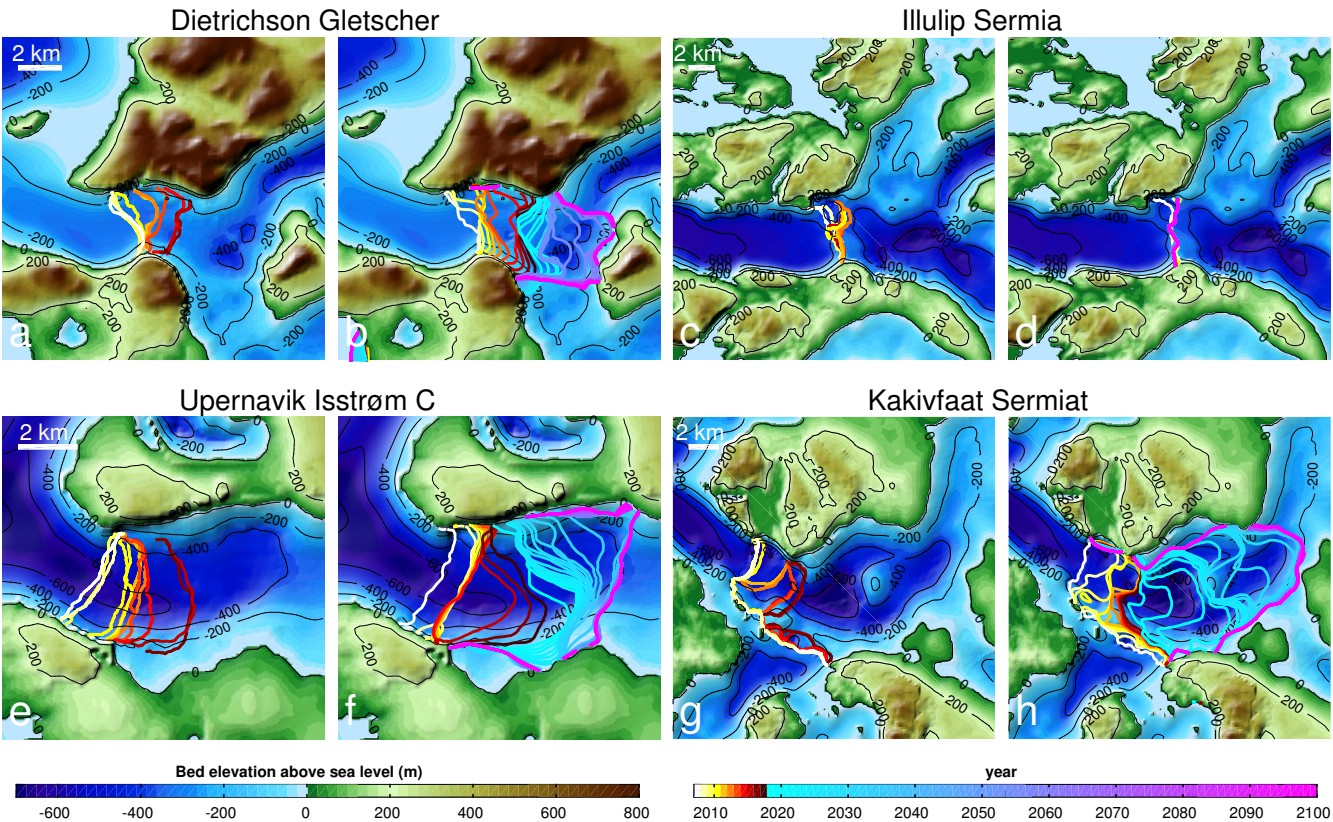

**Figure 4.** Observed (left) and modeled (right) ice front position for Dietrichson Gletscher (a and b), Illulip Sermia (c and d), Upernavik Isstrøm C (e and f), Kakivfaat Sermiat (g and h). under current conditions ($\tilde{T}$ +0°C, $q_{sg} \times 1$). Warm colors are for 2007 to 2017 and cold colors are the model projections for 2017 to 2100.

**Table 1.** Observed and modeled ice front retreat (in km along a centerline) between 2007 and 2017 under current forcing (first two columns), modeled retreat between 2007 and 2030 and between 2007 and 2100, under different scenarios of ocean forcing with today's $q_{sg}$ for individual glaciers along the northwest coast. A more complete table is provided in supplementary material.

| Glacier name | 2017 Retreat (km) | | 2030 Modeled retreat (km) | | | | 2100 Modeled retreat (km) | | | |
|---|---|---|---|---|---|---|---|---|---|---|
| | Observed | Modeled | +0°C | +1°C | +2°C | +3°C | +0°C | +1°C | +2°C | +3°C |
| Sverdrup Gletscher | 2.89 | 2.89 | 8.0 | 12.9 | 13.3 | 14.8 | 13 | 13.9 | 23.4 | 23.4 |
| Dietrichson Gletscher | 3.56 | 3.74 | 4.9 | 7.0 | 8.1 | 13.4 | 6.2 | 54.7 | 54.7 | 54.7 |
| Steenstrup Gletscher | 1.79 | 1.68 | 1.5 | 29.5 | 33.4 | 36.7 | 4.2 | 37.4 | 37.4 | 37.4 |
| Kjer Gletscher | 6.08 | 6.03 | 28.9 | 32 | 34.5 | 36.3 | 38.7 | 38.7 | 39.4 | 40.5 |
| Hayes Gletscher N | -0.266 | -0.533 | 27.5 | 30.4 | 30.7 | 37.9 | 53.9 | 54.3 | 54.3 | 77.1 |
| Hayes Gletscher | 0.475 | 0.104 | 12.9 | 25.4 | 30 | 30.1 | 30.1 | 31.2 | 41.9 | 53.3 |
| Unnamed south Hayes N | 0.06 | 0.06 | 0.6 | 3.3 | 4.4 | 25.2 | 45.3 | 45.3 | 45.4 | 46.8 |
| Unnamed south Hayes M | -0.28 | 0.13 | 0.2 | 2.1 | 12.1 | 12.1 | 39.9 | 40.3 | 42 | 63.6 |
| Unnamed south Hayes SS | 1.12 | 1.12 | 3.1 | 4.0 | 5.3 | 7.3 | 3.5 | 6.5 | 14 | 65.4 |
| Alison Gletscher | 2.36 | 2.64 | 9.5 | 9.8 | 10.5 | 10.6 | 10.5 | 10.5 | 14.5 | 18.3 |
| Illullip Sermia | 0.12 | 0.12 | 0 | 0.9 | 4.6 | 9.5 | 0 | 1.4 | 17.1 | 16.9 |
| Cornell Gletscher | 0.807 | 1.43 | 2.3 | 2.3 | 2.5 | 2.6 | 2.3 | 2.4 | 2.8 | 6.5 |
| Ussing Bræer N | -0.282 | 2.91 | 2.9 | 3.1 | 3.4 | 3.4 | 3.3 | 3.4 | 3.4 | 3.5 |
| Ussing Bræer | 0 | 0 | 0 | 0.1 | 2.3 | 4.54 | 0 | 2.2 | 8.4 | 15.1 |
| Qeqertarsuup Sermia | 0.162 | 0.162 | 0.2 | 0.3 | 1.1 | 2.2 | 0.2 | 0.9 | 4.1 | 9.6 |
| Kakivfaat Sermiat | 4.8 | 4.27 | 12.8 | 19.1 | 19.3 | 19.5 | 19.4 | 19.4 | 19.5 | 19.5 |
| Upernavik Isstrøm N | 0.813 | 0.603 | 4.5 | 4.5 | 5.6 | 10.4 | 4.3 | 4.5 | 5.0 | 11.2 |
| Upernavik Isstrøm C | 2.93 | 2.93 | 4.5 | 6.3 | 8.4 | 8.4 | 6.3 | 7.7 | 8.8 | 15.1 |
| Upernavik Isstrøm S | 0.105 | 0.105 | 0.1 | 5.0 | 10.1 | 13.8 | 0.1 | 17.6 | 27.2 | 29.1 |

If we now look at projections, table 1 and the supplementary table list the modeled retreated distance compared to the 2007 position for all 40 experiments along a central flow line, and figure 5 shows velocity profiles for the different experiments in 2030. Under today's oceanic conditions ($\tilde{T}$ +0°C and $q_{sg} \times 1$), Sverdrup Gletscher is predicted to continue to retreat for another 5 km (i.e., 8 km upstream of its 2007 position) by 2030 and another 5 km by 2100. Under the strongest scenario (i.e. $\tilde{T}$ +3°C and $q_{sg} \times 10$), Sverdrup Gletscher retreats by 23 km compared to 2007 by 2030 and remains there until the end of the century. We find that Sverdrup Gletscher has three distinct stable positions: ~8, 13 and 23 km upstream of the 2007 terminus are ice front positions that we find for a majority of simulations, and they coincide with clear features in the bed topography. Further south, Dietrichson Gletscher will retreat another 1–3 km under the current thermal forcing, and may retreat by up to 55 km by 2100 compared to 2007 if $\tilde{T}$ increases by 1°C or more, or if the subglacial discharge increases by a factor 8 or more. Again, we find clear common retreated positions, 5, 8, 30, 38 and 55 km upstream of the 2007 position, which coincide with

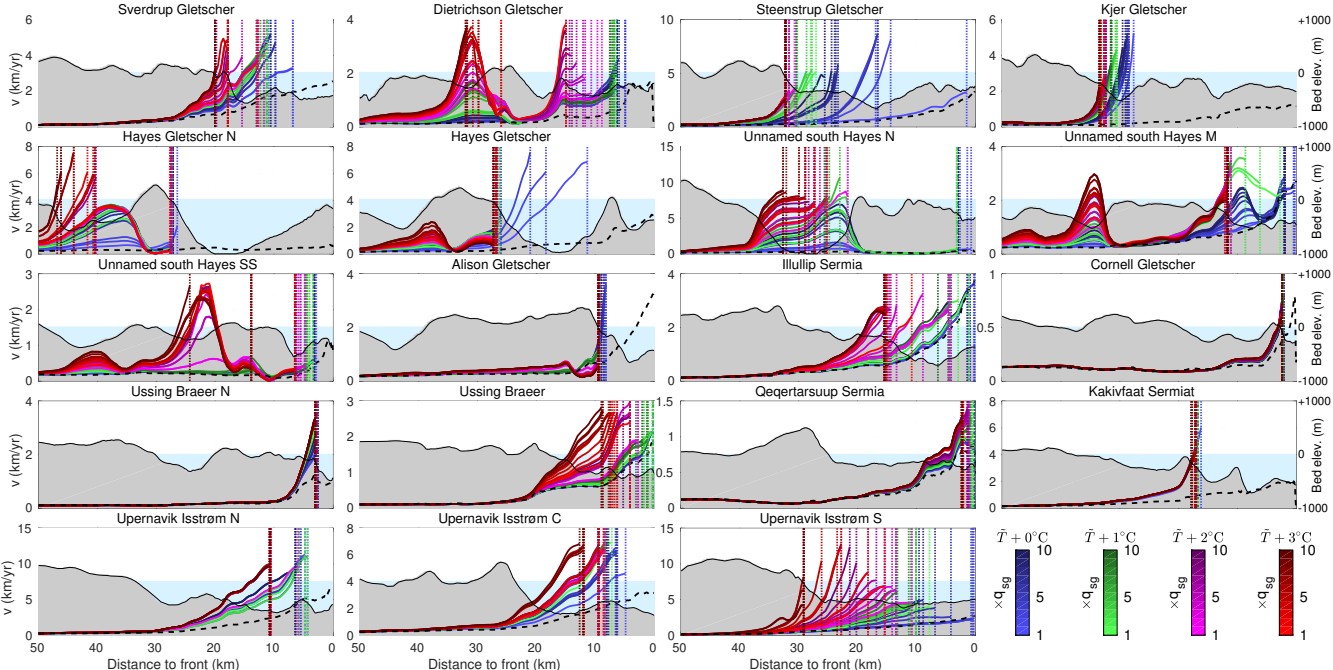

**Figure 5.** Modeled ice velocities (solid lines) and ice front positions (dashed vertical lines) in 2030 for all 40 scenarios. The black dashed line is the current ice velocity (m) and the x axis shows the distance to the current calving front position

topographic features in the bed. Steenstrup Gletscher remains somewhat stable without further ocean warming, but retreats by more than 30 km upstream, where the bed rises above sea level, if the ocean temperature warms by one degree or more, or if the subglacial discharge is doubled. Kjer Gletscher exhibits almost the same behavior for all scenarios: it will continue to retreat another ~40 km upstream over the coming two decades in a region of prograde bed slope, and remain stable there. Hayes

Gletscher N slightly readvanced over the past 10 years but the model suggests that it will retreat by up to 70 km upstream, to where the bed is higher than sea level. Hayes Gletscher would retreat 13 km by 2030, in a marked overdeepening of the bed, and continues to retreat another 17 km to reach a position 30 km upstream of its 2007 position by the end of the simulation. If the thermal forcing increases by 2 or 3°C, the glacier retreats 20 km further inland. The different branches of Unnamed south Hayes also retreat, the Northern branch retreats 45 km by 2100 in all scenarios, to reach a position where the bed rises above

sea level. The middle branch (M) retreats by about 40 km by the end of the century in all cases except if the thermal forcing increases by +3°C, in which case its ice front retreats by 64 km by 2100. The southern branch shows a more binary behavior: it retreats another 3–7 km, depending on the warming scenario, but for enhanced thermal forcing simulations, it may retreat 43 km upstream or even 65 km upstream in the case of a +3°C warming in $\tilde{T}$. Alison Gletscher has been retreating by 2.5 km over the past 10 years, and the model projects that by 2030, in all cases, it will retreat another 7–8 km upstream due to the lack of

features in the bed topography that may stop the retreat. By 2100, the glacier may retreat another 5 km if the thermal forcing increases by +2°C or more.

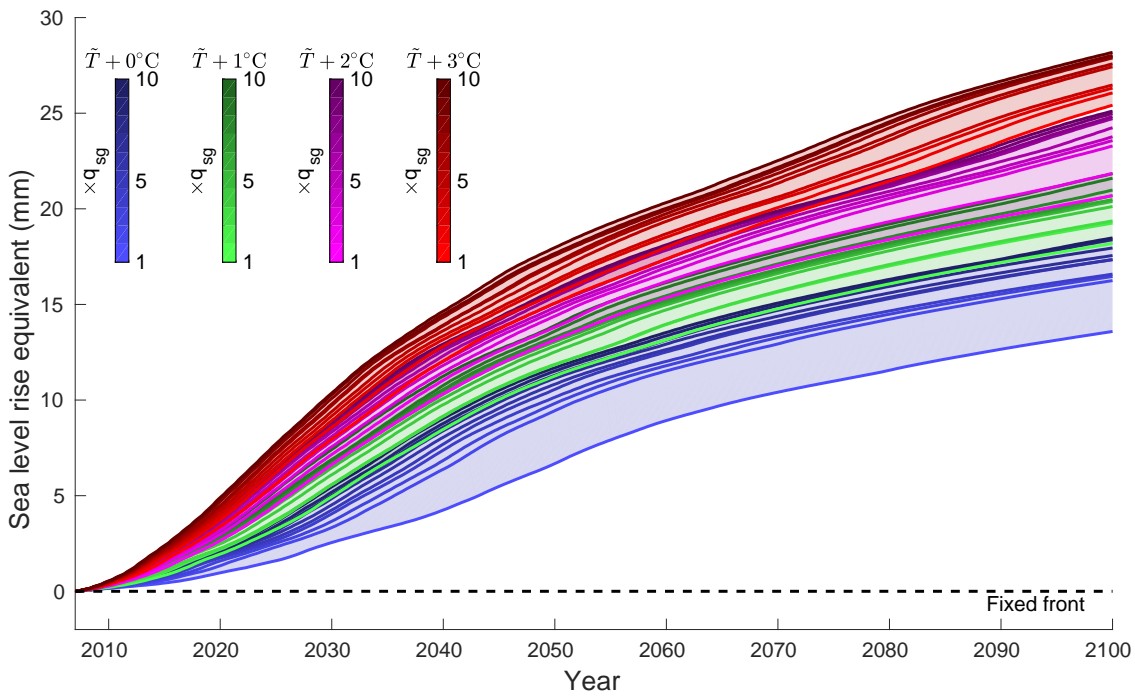

**Figure 6.** Contribution to sea level rise (mm) for all 40 scenarios. The black dashed line is the modeled contribution to sea level with a fixed calving front. All simulations rely on a constant surface mass balance.

Illullip Sermia also has a binary behavior. For the strongest forcing, it retreats by 17–18 km, but in the more conservative scenarios, it stays at its current position that coincides with a large bump in the bed topography. Cornell Gletscher is one of the most stable glaciers of the model: under all scenarios, it retreats another kilometer upstream of its 2017 position and remains stable there, except in the case of +3°C increase in $\tilde{T}$, for which it could retreat by another $\sim$10 km.

5     Ussing Bræer N is the glacier for which we do not capture the advance, but under all scenarios, the model projects that it will remain stable 3 km upstream of its current position, where the bed is very shallow. Ussing Bræer has been stable over the past 10 years, and the model suggests that it may retreat by 9 to 15 km if the ocean thermal forcing increases by 2 to 3°C, but the glacier does not retreat even when the subglacial discharge is multiplied by 10 in the case of no additional increase in $\tilde{T}$. Qeqertarsuup Sermia is also one of the stable glaciers of this region: the model marginally retreats, and under the strongest

10    forcing (+3°C) retreats by about 10 km. Kakivfaat Sermiat, on the other hand, has retreated more than 4 km since 2007. The model suggests that, in all cases, it will retreat another 15 km, where a pronounced feature in the bed topography keeps the ice front stable (figure 5). Our simulations suggest that the glacier may reach this position by 2030 and remain stable there. Upernavik Isstrøm N retreats by 4 km or 11 km depending on the forcing, by 2100. Upernavik Isstrøm C continues to retreat about 3–6 km upstream of its 2007 position, except in the case of a +3°C ocean warming under which it would retreat by 23

km. Finally, Upernavik Isstrøm S would remain stable if the current conditions of $q_{sg}$ and $\tilde{T}$ are maintained, but may retreat between 17 and 29 km if the subglacial discharge is multiplied by a factor of six or if the thermal forcing increases.

Figure 6 shows the contribution to sea level rise of the entire domain for the 40 different scenarios. In all cases, even under current conditions, our simulations suggest that this region will continue to lose mass. The mass loss is significantly higher than in the control experiment, in which we kept the ice front fixed. We also notice that the spread in mass loss due to temperature change (with a fixed $q_{sg}$) is significantly larger than the spread in mass loss due to an increase in subglacial discharge (with fixed $\tilde{T}$). Note that we rely here on a 1960-1991 average surface mass balance, and the projections of ice loss do not account for the increase in surface melt. Our simulations are therefore conservative and should not be used as actual projections.

## 4 Discussion

Our simulations suggest that all glaciers of the northwest coast, except for four (Illullip Sermia, Ussing Bræer, Qeqertarsuup Sermia and Upernavik Isstrøm S) will continue to retreat several kilometers inland under today's thermal forcing and subglacial discharge. Under these conditions, we do not find any glacier which advances.

In all scenarios, we find that the rate and extent of ice front retreat is strongly dependent on the bed topography: ice fronts are stable on topographic bumps and pro-grade bed slopes, and unstable on retrograde bed slope, which is consistent with previous studies (e.g. Warren, 1991; Bassis, 2013; Carr et al., 2015; Catania et al., 2018; Wood et al., 2018). This is for example illustrated in figure 4h, where the ice front jumps from basal bump to basal bump and retreats rapidly in overdeepenings. We find this behavior common to all glaciers in the model domain. There is, however, no "intuitive" way to predict where the glaciers will stabilize without running a model. In most cases, the fjords are not symmetrical or ridges do not go all the way across the fjord walls, which makes it difficult to determine whether the ice front will stabilize or not.

We find that some glaciers, such as Alison Gletscher or Upernavik Isstrøm S, are more sensitive to small increases in ocean thermal forcing, while others, such as Cornell Gletscher or Qeqertarsuup Sermia, are very difficult to destabilize, even under a +3°C increase in ocean thermal forcing. On the other hand, we find that Hayes Gletscher retreats more than 30 km inland into a deep trough once it goes past a ridge, and its velocity increases by a factor of 3 over only 23 years, before restabilizing, under all warming scenarios.

We show here that calving dynamics is an important control on the ice sheet mass balance that should not be ignored. It has been driving the recent dynamic thinning of several Greenland outlet glaciers (e.g. Nick et al., 2009, 2013; Khan et al., 2014; Felikson et al., 2017; Bondzio et al., 2017), and our model study shows that it may continue to control the mass balance of Greenland. Figure 6 shows, for example, that in all cases the system loses a significant amount of mass, and this mass loss is not captured by the model that keeps a fixed calving front. Models keeping ice boundary fixed (e.g., Gillet-Chaulet et al., 2012; Seroussi et al., 2013; Bindschadler et al., 2013) will consistently under-estimate ice sheet mass loss as they do not capture the effect of ocean warming. These conservative projections should therefore be treated with caution and efforts should be made to include moving boundaries in continental scale simulations of the Greenland ice sheet in order to account for ice-ocean interactions, despite the complexity and high mesh/grid resolution needed to resolve moving boundaries (∼1 km, Bondzio

et al., 2016) of such simulations. It is also important to note that the future evolution of Greenland is strongly influenced by the ocean (through the ocean thermal forcing). It is important to not only force predictive ice sheet models with projections of surface mass balance, but also to include projections of ocean thermal forcing at the fjord mouth. There may also be some positive or negative feedbacks between changes in surface mass balance and calving. More surface melt, for example, could enhance calving through hydrofracture, while at the same time reducing the ice thickness at the calving front, hence reducing the stress. Ideally, the community should move towards ice-ocean-climate coupled models to fully understand the processes that control the stability of the ice sheet (Nowicki and Seroussi, 2018).

Another interesting aspect of this analysis is that glaciers are more sensitive to an increase of one to two degrees in ocean thermal forcing than in a 5 to 10-fold increase in subglacial discharge. This is actually a result of the parameterization of undercutting used here (eq. 3), which is itself more sensitive to $\tilde{T}$ than $q_{sg}$: the parameterization is sub-linearly dependent on $q_{sg}$ and above-linear in $\tilde{T}$. The effect of surface runoff is also limited to summer months, while the ocean thermal forcing affects the glacier year-round. That being said, we do not account for other effects that surface runoff may have on ice dynamics, such as enhanced damage due to hydrofracture, which may lead to a decrease in the stress threshold $\sigma_{\mathrm{max}}$. Glaciers might therefore be more prone to retreat as $q_{sg}$ increases than what is captured by the current model.

Among other limitations in this study, no numerical ocean model is included: the thermal forcing is prescribed and dictates the rate of undercutting. Similarly, the calving law is not capturing all the modes of calving and requires more validation. This study is indeed relying on two parameterizations that drive the response of the model to ocean forcings. It is therefore critical to further validate these parameterizations, or develop new ones that include more physics and better capture the transfer of heat from the fjord mouth to the calving face, and iceberg calving. We also assumed that the subglacial discharge was distributed uniformly across the calving front but observations show that the majority of discharge is routed to one or more large channel outlets (e.g., Fried et al., 2015). Frontal undercutting is therefore not distributed uniformly either, even though numerical experiments suggest that the uncertainty in melt is on the order of 15% (Xu et al., 2013). We have also shown how our results were strongly influenced by the bed topography. While the bed is pretty well constrained in this region (Morlighem et al., 2017), it is not free of error, and we have shown again here how important features in the bed topography are for calving front stability.

More importantly, this study paves the way for a Greenland-wide projection that includes realistic parameterizations of moving boundaries, which will provide more reliable estimates than current models that do not include calving. This work also suggests that development of more accurate parameterization of undercutting and calving should be developed as they control the response of the model, and its stability in future scenarios. While this work is a first step in this direction, more validation should be performed on these parameterizations, and future parameterizations of undercutting and calving will make models more reliable.

## 5    Conclusions

In this study, we modeled the response of the northwest coast of Greenland to enhanced oceanic forcing and subglacial discharge and found that this sector will continue to lose mass over the coming decades, regardless of the scenario adopted. The model confirms that ice-ocean interactions have the potential to trigger extensive glacier retreat over a short amount of time (i.e. decades), but the bed topography controls the magnitude and rate of retreat. Overall, the model showed greater sensitivity to enhanced thermal forcing compared to subglacial discharge, but did not account for other effects that runoff may have on ice flow. While more work on validating this parameterization of undercutting and the calving law employed here is needed, we showed that accounting for ice front dynamics can lead to significantly more ice loss than with a fixed calving front. Under the current oceanic and atmospheric conditions, this sector alone will contribute more than 1 cm to sea level rise by the end of this century, and up to 3 cm in the worst-case scenario.

*Code and data availability.*   The data used in this study are freely available on the National Snow and Ice Data Center, or upon request to the authors. ISSM is open source and freely available at http://issm.jpl.nasa.gov.

*Author contributions.*   MM set up the model, designed the experiments, ran the simulations and wrote the manuscript. MW provided the data required to compute the rate of undercutting, HS and YC assisted in conducting the numerical experiments. All authors participated in the writing of the manuscript.

*Competing interests.*   None

*Acknowledgements.*   This work was performed at the University of California Irvine under a contract with the National Aeronautics and Space Administration, Cryospheric Sciences Program (#NNX15AD55G), and the National Science Foundation's ARCSS program (#1504230). Resources supporting this work were provided by the NASA High-End Computing (HEC) Program through the NASA Advanced Supercomputing (NAS) Division at Ames Research Center. This work would not have been possible without data from NASA Ocean Melting Greenland EVS-3 mission, and NASA Operation IceBridge. We thank Andy Aschwanden, an anonymous reviewer and the editor Andreas Vieli for their helpful and insightful comments.

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
