# Peer review of "Modeling the response of Northwest Greenland to enhanced ocean thermal forcing and subglacial discharge"

_The Cryosphere, 2018_

## Referee Comment (RC1) · A. Aschwanden (Referee) · 1 Nov 2018

This is an important and relevant paper as it extends previous efforts by the same group from a single outlet glacier to a regional view. It certainly deserves publication after some polishing of the text. While the science is sound, the writing is relatively poor and sloppy, with many typos and grammatical errors. It seems the manuscript was put together in a haste and would have benefited significantly from a round of proof reading before submission (see all my technical comments).

Cheers, Andy Aschwanden

[Figure]

**Methods**

The methods and data section needs polishing and clarification: Please explain more carefully how subglacial discharge and thermal forcing are applied, are these daily or monthly forcing, or annually averaged? Is the subglacial discharge averaged over a certain time period like the surface mass balance? What is the resolution and the time step of the model? Since it's an unstructured grid, please inform the reader of the minimum and maximum cell size.

Equation 2 uses ambiguous notation First, TF should not be used as a variable because it could mean $TxF$, how about something like $T_h$?. I realize that this kind of sloppy notation has become more widespread in the glaciological literature over the past few years, and that the authors want to use the same notation as previous publications. Second, it took me several readings to understand that $q_{sg} \times 1$ and TH $+1°C$ is a shorthand for anomalies. The problem with this is that it is unclear when the authors talk about the initial (present day) forcing, and when anomalies are meant. I think what the authors are doing is something like this

$$\dot{M} = (Ah(q_{sg}(x,y)q_a)^\alpha(t) + B)(T_h(x,y) + T_a(t))^\beta$$

, (1)

where $q_a$(t) and $T_a(t)$ are multiplicative and additive time-dependent scalar anomalies, respectively. Use of a notation like this would improve clarity.

Regarding climate (surface mass balance) forcing: Why do you use the 1960-1991 average surface mass balance? This could possibly effect both the calibration and the projections. The 1960-1991 average was longer than today, thus to match the observed frontal retreat, your calibration procedure for the ocean forcing will have to compensate. Furthermore, use of the 1960-1991 average SMB for projections is questionable and

as a consequence, one has little confidence in the sea-level contribution (Figure 5). As the focus of this paper is on glacier front retreat, I wonder if I'd be best to remove Figure 5 (and related text)? I do not think the manuscript would lose anything.

Detailed comments

p 1, l 8: Northwest -> northwest

p 1, l 13-14: "While these parametrizations are approximations..." this statement is almost universally true and I thus suggest to remove it from the abstract with any loss. How about "These parametrizations have shown to provide reliable estimates..."

p 1, l 17: include the year. The 50km retreat occurs from present day until year 2100, otherwise the reader might think the glacier retreats 50km over the course of 15 years.

p 2, l 9: remove comma. "...the rate of undercutting at the calving face..."

p 2, l 11-12. Rephrase "We don't...", this sentence does not make much sense to. Or leave the sentence out?

p 2, l 20-21: It remains unclear, however, to which extent glaciers of the...

p 2, l 30-31: "While a lot of progress has been made in terms of capturing ice flow through the development of new, higher-order stress balance solvers, ..." I respectfully disagree with this statement; significant progress was due to the availability of more accurate ice thickness instead. I'm not aware of a publication that demonstrates that higher-order stress balance solvers have greatly improved our ability to capture ice flow on a continental scale.

p 3, l 25: insert comma after equation

p 3, l 27-28: A lot of research is currently being dedicated to derive parametrizations for $c$ and $\dot{M}$; here we chose to recent parametrizations described below

p 5, l 4: insert comma after equation

p 6, l 1 simplification,but -> simplification, but

p 6-7: "As we do not run a coupled model, we rely on the last year of constrained rate of undercutting (year 2016) and repeat it" This sentence does not make sense. As I understand it, you calculate undercutting from thermal forcing and subglacial discharge, what do you mean with "repeating"?

p 8, l 13: overestimates the retreat on the southern...

p 8, l 31: Kjer Gletscher exhibits almost the same...

p 8, l 33-34: I think it should read "up to 70km upstream to where the bed..." (not sure though)

p 8, l 34: add year: but continue to retreat another 17km by 2100 to reach...

p. 9, l 1: the northern branch retreats 45km...

p. 9, l 9: "has" is very colloquial. Use "shows" or "exhibits" instead.

p. 9-10: "In our simulations, Cornell Gletscher shows some of the most stable behavior of all investigated glaciers: under all scenarios, it retreats roughly another kilometer upstream." Remove the "or so", this is too colloquial.

p. 10, l 4: the model projects that...

p. 10, l 6: ...no additional increase in TF

p. 10, l 8: I think it should read "..., on the other hand, has retreated more..."

p. 10, l 10: Our simulations suggest that the glacier may reach..."

p. 10, l 11: clarify "by 4km or 11km", on what does this depend?

p. 11, l 2: is multiplied by a factor of six

p. 11, l 5: in the control experiment, in which we kept the ice front fixed.

[Figure]

p. 11, l 11: Under these conditions, ...

p. 12, l 12: "(not shown here) -> this results is highlighted in the abstract, I thus think it needs to be shown here.

p. 12, l 24: move towards coupled ice-ocean-climate models

p. 12, l 33: "Among other limitations...". Clarify and rewrite. "the thermal forcing is dictated by the undercutting" ? Isn't it the other way round?

p 5, l 13-14 and 22: is there a contradiction? First you say you are using ECCO from 1992-2015 and further down it's 2007 until 2015? I understand that the simulations start in 2007, so what is the ECCO data prior to 2007 used for?

Figures: the figures are beautiful.

Figure 1: ..., and white crosses indicate the locations of CTD data from NASA's Oceans Melting Greenland campaign that were used to calibrate thermal forcing

Figure 3: add units to colorbars.

---

## Referee Comment (RC2) · Anonymous Referee #2 · 20 Dec 2018

General comments:

The authors explore the sensitivity of Northwest Greenland's marine-terminating glaciers to decadal-scale increases in thermal forcing and subglacial discharge. Using the Ice Sheet System Model (ISSM), they run an ensemble of 21st century experiments with thermal forcing increasing by up to 3 deg C, and subglacial discharge increasing up to a factor of 10. The model uses two parameterizations that determine the terminus location: one for calving, driven by tensile stresses, and the other for undercutting, driven by thermal forcing and subglacial discharge. It makes innovative use of ECCO ocean output, along with new bed topography data from NASA. The authors find a wide

range of glacier responses, with some glaciers sensitive to small increases in thermal forcing, and others quite stable. They argue that bed topography controls the rate and magnitude of retreat. The paper is clearly structured. It places the problem in scientific context, lays out methods and parameterizations, quantifies the results, draws general conclusions, and discusses model limitations. The experiments are a significant step toward Greenland-wide projections of the evolution of Greenland's marine outlet glaciers. However, some sections are written in a cursory way without enough details and justification. In particular, the paper seems to rely on some implicit assumptions that are not fully explained and defended, thus casting doubt on the validity of the model calibration. Although the study is timely and important, the methodology and description should be improved, as described below.

Specific comments:

First, I will restate what seem to be the underlying assumptions in Section 2: The terminus location of marine-terminating glaciers (at least in Northwest Greenland) is determined mainly by (1) mass transport; (2) undercutting driven by thermal forcing (TF) and subglacial discharge, as quantified by Eq. 2; and (3) calving proportional to ice velocity and tensile stress, as described by Eq. 3. The steady-state terminus location is determined by a balance between (1), which advances the front, and (2) and (3), which drive frontal retreat. Processes (2) and (3) are largely independent of each other. Marine glacier retreat of the past decade can be attributed primarily to increased thermal forcing and undercutting.

One way to test the validity of these assumptions would be to calibrate the model by fitting simulated termini to observed locations prior to retreat. The model could be initialized using observed or model-derived values of velocity, tensile stress, runoff-derived discharge, and ocean thermal forcing, appropriate for a period before the recent retreat (say, late 20th century) when the termini were relatively stable. Of these fields, TF might be the least certain (as suggested on p. 7, l. 2), so one approach to initialization would be to invert for TF based on pre-retreat terminus locations. This would give a
baseline TF to which anomalies would be added.

In this study, however, there is no calibration based on pre-retreat termini. If I understand Section 2 correctly, the model is initialized to 2007 geometry and then run forward with a linear sliding law and RACMO-derived runoff, along with the undercutting and calving parameterizations of eq. 2 and 3. The calving parameter sigma_max is adjusted in each basin to match the observed retreat of the past decade.

This approach left me wondering how much of the simulated decadal-scale retreat is associated with recent increases in TF (as shown in Fig. 2b for ECCO). Part of the retreat could be a model transient that would occur without increasing TF, because of biases in SMB, basal friction, or other factors. Based on information given in the paper, I don't know how to make this judgment, and to be confident that eq. 2 and 3 capture the essential physics (albeit with uncertainty in empirical parameters).

If it is not feasible to calibrate the model based on pre-retreat terminus locations, I would ask the modelers to describe the difficulties and explain why their approach is preferred.

Comments with page and line references follow below.

p. 1: The abstract is longer than necessary. Some of the details and elaborations could be left out (for example, the sentence beginning "While these parameterizations remain approximations...").

p. 2, l. 11: The last sentence of this paragraph might fit better at the end of the Introduction section.

p. 3, l. 10: In this paragraph, please state the ISSM grid resolution (or range of resolutions).

p. 3, l. 17: It would be helpful to see an equation for the Budd sliding parameterization, along with the chosen parameter values (such as a sliding proportionality constant C). Using this parameterization, how close is the fit of simulated glacier velocities to

Interactive
comment

observed velocities?

p. 3, l. 23: Since undercutting is closely related to processes that might be classified as calving, it would be helpful to state that calving and undercutting are considered to be independent processes for purposes of the paper.

p. 3, l. 29: For readers not familiar with the Rignot et al. paper, please describe the motivation for choosing this particular functional form for undercutting, and these parameter values. For example, why is the B term needed? Are there theoretical reasons to expect alpha and beta to have roughly these values, or are they strictly empirical? Why the dependence on h?

p. 4, Fig. 1: This is a very useful and visually attractive figure.

p. 5, l. 1: Please give equations showing how TF is computed.

p. 5, ll. 9ff: This is a helpful explanation of the interplay between bed topography and thermal forcing.

p. 5, Fig. 2: This is another useful figure, which helped me visualize the model forcing, but the caption is not very informative. For instance, the caption for Fig 2a could state that the initial calving front is at x = 0 km and the fjord mouth is at x = 80 km (if I'm interpreting it correctly). For Fig. 2b, please give the source of the data.

p. 5, l. 17: "repeat TF of year 2016 until the end of the simulation." I found this confusing. Do you mean that TF from 2016 is the baseline to which the TF anomaly is added?

p. 6, l. 9: Please see the above comments on model calibration. There is no explanation here of why sigma_max is the specific parameter chosen for calibration, or of why calibration to the observed retreat is preferable to calibration to pre-retreat terminus location.

p. 6, l. 10: Landsat-derived ice front retreat is mentioned here for the first time. I

suggest describing the observed retreat, perhaps with a reference to the left column of numbers in Table 1, as part of the background discussion.

p. 6, l. 16: Can you say approximately how many CMIP5 models were used to compute this average anomaly, and what is the spread among models?

p. 6, l. 17: I'm not sure the 2-degree Paris target is entirely relevant here, given that the target might be exceeded, and we want to know the consequences of missing the target. It would be more appropriate to choose an upper limit based on the CMIP5 spread.

p. 6, l. 24: Can you say why you increase the TF anomaly instantly, instead of phasing it in linearly as might be more realistic? Similarly for SMB/discharge.

p. 6, l. 26: It is unclear exactly what is being repeated here.

p. 7, ll. 3-5: Referring again to the comments above, I'm wondering if a bias correction to the ECCO data would be more defensible than adjusting sigma_max. Then you would more likely be capturing the recent retreat for the right reason.

p. 8, l. 2: How informative is it that ice front retreat is in good agreement with observations when sigma_max is tuned? Does this suggest that there is something fundamentally "right" about the parameterizations, or does it simply reflect high sensitivity to modest changes in sigma_max?

p. 8, l. 9: The text states that the model overestimates the recent retreat of Kakivfaat Sermiat, but Table 1 shows a small underestimate.

p. 10, Fig. 5: This figure nicely illustrates that the TF anomaly is the primary driver of retreat. However, I wondering how the shape of the figure would be different if anomalies were ramped up gradually. Also, I suggest that the caption briefly state what is left out of the fixed-front simulation, such as SMB changes.

p. 11, l. 17: I couldn't find a description of the control experiment in which the ice front

is held fixed.

p. 12, ll. 3ff: The paper makes a strong case for the important of calving dynamics. At the same time, it does not quantify mass changes due to a more negative SMB, or discuss possible feedbacks of SMB on dynamics. For instance, could a decreasing SMB potentially cause much more mass loss than dynamic retreat? Could SMB-driven thinning significantly modify the calving dynamics (e.g., through reduced tensile stresses)? I realize that SMB changes are beyond the scope of the modeling study, but it would be helpful to talk about them in Section 4.

p. 12, l. 7: This statement about models with fixed calving fronts seems too general. For example, consider a model without a physically based calving law, in which calving is simply prescribed at the present-day CF. Suppose the model is forced with increasingly negative SMB. This could result in significant thinning, and perhaps ungrounding, of ice all the way to the CF, without necessarily moving the CF. This isn't to say that moving boundaries aren't an improvement, but just to acknowledge that models without moving boundaries may still be able to make useful projections, which might not be overly conservative (especially if SMB dominates the mass balance).

p. 12, l. 15: I agree that this is an interesting result, which might not have been guessed ahead of time. Given the result, it would be helpful to comment (either here or in Section 2) on the robustness or theoretical justification of alpha and beta.

Technical corrections:

p. 2, l. 1: Please be consistent in capitalization of "Northwest" vs. "northwest"

p. 2, l. 11: don't -> do not

p. 2, l. 19: Maybe "on the edge" -> "on the verge"

p. 2, l. 27: To me, "plan-view" suggests 2D in the xy plane. Maybe "3D"?

p. 2, l. 28: "a lot of" -> "much". Also p. 3, l. 24.

P. 2, l. 33: Define RCP

Fig. 1 caption: "are used to calibrate the thermal forcing." Also, capitalize "south" in the figure.

p. 5, l. 6: Maybe reword as "...the assumption of uniformly distributed melt generates only..."

p. 5, l. 11: As worded, the subject of "decreases" is "calculation", which isn't intended. Maybe change to "The calculated effective depth"

p. 5, l. 17: "future simulation" -> "simulations of future climate"

p. 6, l. 7: Hyphenate "real-world"

p. 7, Fig. 3 caption: undercutting rate should be m/day instead of m/yr?

p. 7, l. 7: 4 -> four

p. 9, l. 6: "about 8, 13 and 23 km upstream are distances we find..."; awkward wording.

p. 9, l. 9: position -> positions

p. 9, l. 11: "under no warming condition" -> "without further ocean warming"

p. 10, l. 5: Run-on sentence

p. 11, l. 2: "10 km or so" -> "~ 10 km"

p. 11, l. 3: project -> projects

p. 11, l. 23: advances -> advance

p. 11, l. 25: retrograde (no hyphen)

p. 12, l. 17: sensitive -> more sensitive

---

## Author Comment (AC1) · 18 Jan 2019

**Modeling the response of Northwest Greenland to enhanced ocean thermal forcing and subglacial discharge**
**– Response to reviewers –**

Mathieu MORLIGHEM et al.

January 17, 2019

We thank Andy Aschwanden and an anonymous reviewer for their very constructive and insightful comments. We address their remarks below point by point.

**1   Reviewer #1: Andy Aschwanden**

*This is an important and relevant paper as it extends previous efforts by the same group from a single outlet glacier to a regional view. It certainly deserves publication after some polishing of the text. While the science is sound, the writing is relatively poor and sloppy, with many typos and grammatical errors. It seems the manuscript was put together in a haste and would have benefited significantly from a round of proof reading before submission (see all my technical comments).*

We thank the reviewer for his comment, and apologize about the typos that were in the manuscript, despite the proofreading from all authors before the initial submission.

*The methods and data section need polishing and clarification: Please explain more carefully how subglacial discharge and thermal forcing are applied, are these daily or monthly forcing, or annually averaged? Is the subglacial discharge averaged over a certain time period like the surface mass balance?*

They are both monthly averaged, we added this to the main text.

*What is the resolution and the time step of the model? Since it's an unstructured grid, please inform the reader of the minimum and maximum cell size.*

Done (between 100 m and 1 km and 7-day time step)

*Equation 2 uses ambiguous notation. First, TF should not be used as a variable because it could mean TxF, how about something like $T_h$? I realize that this kind of sloppy notation has become more widespread in the glaciological literature over the past few years, and that the authors want to use the same notation as previous publications.*

We indeed tried to use existing notations. We replated $TF$ by $\tilde{T}$.

*Second, it took me several readings to understand that $q_{sg} \times 1$ and TF +1°C is a shorthand for anomalies. The problem with this is that it is unclear when the authors talk about the initial (present day) forcing, and when anomalies are meant. I think what the authors are doing is something like this:*

$$\dot{M} = (Ah(q_{sg}(x,y)q_a)^\alpha(t) + B)(T_h(x,y) + T_a(t))^\beta, \qquad (1)$$

*where $q_a(t)$ and $T_a(t)$ are multiplicative and additive time-dependent scalar anomalies, respectively. Use of a notation like this would improve clarity.*

This is an excellent point, we added this equation to the manuscript in the "Experiments" section and discuss anomalies accordingly.

*Regarding climate (surface mass balance) forcing: Why do you use the 1960-1991 average surface mass balance? This could possibly effect both the calibration and the projections. The 1960-1991 average was longer than today, thus to match the observed frontal retreat, your calibration procedure for the ocean forcing will have to compensate. Furthermore, use of the 1960-1991 average SMB for projections is questionable and as a consequence, one has little confidence in the sea-level contribution (Figure 5). As the focus of this paper is on glacier front retreat, I wonder if I'd be best to remove Figure 5 (and related text)? I do not think the manuscript would lose anything.*

We decided to use the 1960-1991 average surface mass balance as this was a period for which the Greenland ice sheet was approximately in equilibrium and we wanted to conduct here a sensitivity study with respect to ice front dynamics. We therefore want to isolate the frontal forcing. The simulations shown here are not projections, but compare the effect of an increase in TF or $q_{sg}$ on ice front dynamics, and how this translates to mass loss. We tried to calibrate the models with different SMB fields and the rate of retreat was not significantly sensitive to the surface mass balance used over the hindcast period. We decided to keep Figure 5, to highlight the fact that ice front dynamics can account for large mass losses, but revised the text slightly to highlight the fact that with projected SMBs, the mass loss would be even greater.

*Detailed comments*

*p 1, l 8: Northwest → northwest*

Done

*p 1, l 13-14: "While these parametrizations are approximations..." this statement is almost universally true and I thus suggest to remove it from the abstract with any loss. How about "These parametrizations have shown to provide reliable estimates..."*

We removed the sentence based on reviewer #2's comments.

*p 1, l 17: include the year. The 50km retreat occurs from present day until year 2100, otherwise the reader might think the glacier retreats 50 km over the course of 15 years.*

Done

*p 2, l 9: remove comma. "...the rate of undercutting at the calving face..."*

Done

*p 2, l 11-12. Rephrase "We don't...", this sentence does not make much sense to. Or leave the sentence out?*

We removed the sentence.

*p 2, l 20-21: It remains unclear, however, to which extent glaciers of the...*

Rephrased.

*p 2, l 30-31: "While a lot of progress has been made in terms of capturing ice flow through the development of new, higher-order stress balance solvers, ..." I respectfully disagree with this statement; significant progress was due to the availability of more accurate ice thickness instead. I'm not aware of a publication that demonstrates that higher-order stress balance solvers have greatly improved our ability to capture ice flow on a continental scale.*

We agree with the reviewer that the improvement in bed topography (and higher mesh/grid resolution) was undoubtebly critical in improving models, as shown by *Aschwanden et al.* [2016]. We used the term "higher-order" a bit loosely here, meaning "non-shallow-ice models". It is known that SIA does not include membrane stresses and therefore does not capture the effect of ice shelf buttressing, or ice shelf collapse, among other important processes. We rephrased the sentence to avoid any ambiguity.

*p 3, l 25: insert comma after equation*

Done

*p 3, l 27-28: A lot of research is currently being dedicated to derive parametrizations for $c$ and $\dot{M}$; here we chose to recent parametrizations described below*

Done

*p 5, l 4: insert comma after equation*

Done

*p 6, l 1 simplification,but $\rightarrow$ simplification, but*

Done

*p 6-7: "As we do not run a coupled model, we rely on the last year of constrained rate of undercutting (year 2016) and repeat it" This sentence does not make sense. As I understand it, you calculate undercutting from thermal forcing and subglacial discharge, what do you mean with "repeating"?*

We repeat the 2016 time series from 2017 onwards for all years until 2100. We clarified the text.

*p 8, l 13: overestimates the retreat on the southern...*

Done

*p 8, l 31: Kjer Gletscher exhibits almost the same...*

Done

*p 8, l 33-34: I think it should read "up to 70 km upstream to where the bed..." (not sure though)*

Done

*p 8, l 34: add year: but continue to retreat another 17km by 2100 to reach...*

Done

*p. 9, l 1: the northern branch retreats 45km...*

Done

*p. 9, l 9: "has" is very colloquial. Use "shows" or "exhibits" instead.*

Done

*p. 9-10: "In our simulations, Cornell Gletscher shows some of the most stable behavior of all investigated glaciers: under all scenarios, it retreats roughly another kilometer upstream." Remove the "or so", this is too colloquial.*

Done

*p. 10, l 4: the model projects that...*

Done

*p. 10, l 6: ...no additional increase in TF*

Done

*p. 10, l 8: I think it should read "..., on the other hand, has retreated more..."*

Done

*p. 10, l 10: Our simulations suggest that the glacier may reach..."*

Done

*p. 10, l 11: clarify "by 4km or 11km", on what does this depend?*

Done

*p. 11, l 2: is multiplied by a factor of six*

Done

*p. 11, l 5: in the control experiment, in which we kept the ice front fixed.*

Done

Done

*p. 12, l 12: "(not shown here) → this results is highlighted in the abstract, I thus think it needs to be shown here.*

This is also something that was requested by reviewer #2. We added an additional figure (Figure 5) that shows velocity profiles for all scenarios in 2030, as well as the ice front position and bed topography. This figure also helps understand the behavior of the glaciers. To be consistent with the figure (that shows the state of the glaciers in year 2030), we changed the acceleration factor to 3 over 23 years.

*p. 12, l 24: move towards coupled ice-ocean-climate models*

Done

*p. 12, l 33: "Among other limitations...". Clarify and rewrite. "the thermal forcing is dictated by the undercutting" ? Isn't it the other way around?*

Yes, this sentence was confusing and has been rephrased.

*p 5, l 13-14 and 22: is there a contradiction? First you say you are using ECCO from 1992-2015 and further down it's 2007 until 2015? I understand that the simulations start in 2007, so what is the ECCO data prior to 2007 used for?*

This is a good point. We actually only use the data from 2007 to 2015, we corrected the manuscript.

*Figures: the figures are beautiful.*

Thank you!

*Figure 1: ..., and white crosses indicate the locations of CTD data from NASA's Oceans Melting Greenland campaign that were used to calibrate thermal forcing*

Done

*Figure 3: add units to colorbars.*

Done

**2 Reviewer #2**

*General comments:*

*The authors explore the sensitivity of Northwest Greenland's marine-terminating glaciers to decadal-scale increases in thermal forcing and subglacial discharge. Using the Ice Sheet System Model (ISSM), they run an ensemble of 21st century experiments with thermal forcing increasing by up to 3 deg C, and subglacial discharge increasing up to a factor of 10. The model uses two parameterizations that determine the terminus location: one for calving, driven by tensile stresses, and the other for undercutting, driven by thermal forcing and subglacial discharge. It makes innovative use of ECCO ocean output, along with new bed topography data from NASA. The authors find a wide range of glacier responses, with some glaciers sensitive to small increases in thermal forcing, and others quite stable. They argue that bed topography controls the rate and magnitude of retreat. The paper is clearly structured. It places the problem in scientific context, lays out methods and parameterizations, quantifies the results, draws general conclusions, and discusses model limitations. The experiments are a significant step toward Greenland-wide projections of the evolution of Greenland's marine outlet glaciers. However, some sections are written in a cursory way without enough details and justification. In particular, the paper seems to rely on some implicit assumptions that are not fully explained and defended, thus casting doubt on the validity of the model calibration. Although the study is timely and important, the methodology and description should be improved, as described below.*

We thank the reviewer for his general assessment and hope that the new version of the manuscript addresses all of the concerns.

*Specific Comments:*

*First, I will restate what seems to be the underlying assumptions in Section 2: The terminus location of marine-terminating glaciers (at least in Northwest Greenland) is determined mainly by (1) mass transport; (2) undercutting driven by thermal forcing (TF) and subglacial discharge, as quantified by Eq. 2; and (3) calving proportional to ice velocity and tensile stress, as described by Eq. 3. The steady-state terminus location is determined by a balance between (1), which advances the front, and (2) and (3), which drive frontal retreat. Processes (2) and (3) are largely independent of each other. Marine glacier retreat of the past decade can be attributed primarily to increased thermal forcing and undercutting.*

*One way to test the validity of these assumptions would be to calibrate the model by fitting simulated termini to observed locations prior to retreat. The model could be initialized using observed or model-derived values of velocity, tensile stress, runoff-derived discharge, and ocean thermal forcing, appropriate for a period before the recent retreat (say, late 20th century) when the termini were relatively stable. Of these fields, TF might be the least certain (as suggested on p. 7, l. 2), so*

*one approach to initialization would be to invert for TF based on pre-retreat terminus locations. This would give a baseline TF to which anomalies would be added. In this study, however, there is no calibration based on pre-retreat termini. If I understand Section 2 correctly, the model is initialized to 2007 geometry and then run forward with a linear sliding law and RACMO-derived runoff, along with the undercutting and calving parameterizations of eq. 2 and 3. The calving parameter sigma_max is adjusted in each basin to match the observed retreat of the past decade. This approach left me wondering how much of the simulated decadal-scale retreat is associated with recent increases in TF (as shown in Fig. 2b for ECCO). Part of the retreat could be a model transient that would occur without increasing TF, because of biases in SMB, basal friction, or other factors. Based on information given in the paper, I don't know how to make this judgment, and to be confident that eq. 2 and 3 capture the essential physics (albeit with uncertainty in empirical parameters). If it is not feasible to calibrate the model based on pre-retreat terminus locations, I would ask the modelers to describe the difficulties and explain why their approach is preferred.*

We actually had a similar idea initially to calibrate the model. There are a few reasons why we ended up not doing it. First, it is virtually impossible to find a surface DEM at the scale of northwest Greenland prior to 2007. There are some regions where we have decent DEMs based on photogrammetry, but their spatial extent is too limited. Another reason is that stable glaciers generally have their terminus on a distinct feature (ledge, ridge, etc) in the bed topography and the numerical model is also stable for a wide range of $\sigma_{max}$. Constraining the calving threshold, $\sigma_{max}$, is easier to do for retreating calving front as we constrain the *rate of retreat* as opposed to just the *stability* of the glacier. This is something that we noticed in *Morlighem et al.* [2016] and *Choi et al.* [2018]. It is also difficult to invert for TF because of its natural variability, while we can assume that $\sigma_{max}$ does not change over short time scales. Ultimately, we agree with the reviewer that the situation is not satisfying given that the projections rely heavily on 2 parameterizations that need further validation, which is what we mention in the discussion and conclusion.

*Comments with page and line references follow below.*

*p. 1: The abstract is longer than necessary. Some of the details and elaborations could be left out (for example, the sentence beginning "While these parameterizations remain approximations. . .").*

We removed this sentence.

*p. 2, l. 11: The last sentence of this paragraph might fit better at the end of the Introduction section.*

We removed this sentence based on reviewer #1's suggestion.

*p. 3, l. 10: In this paragraph, please state the ISSM grid resolution (or range of resolutions).*

Done

*p. 3, l. 17: It would be helpful to see an equation for the Budd sliding parameterization, along with the chosen parameter values (such as a sliding proportionality constant C). Using this parameterization, how close is the fit of simulated glacier velocities to observed velocities?*

Done

*p. 3, l. 23: Since undercutting is closely related to processes that might be classified as calving, it would be helpful to state that calving and undercutting are considered to be independent processes for purposes of the paper.*

This is a good point but we think this is already mentioned in the text (Section *Calving parameterization*: " It is also assumed here that $c$ and $\dot{M}$ are independent, which is a simplification").

*p. 3, l. 29: For readers not familiar with the Rignot et al. paper, please describe the motivation for choosing this particular functional form for undercutting, and these parameter values. For example, why is the B term needed? Are there theoretical reasons to expect alpha and beta to have roughly these values, or are they strictly empirical? Why the dependence on h?*

The coefficients $\alpha$ and $\beta$ are close to that expected from the plume theory [*Jenkins et al.*, 2010; *Jenkins*, 2011], but were determined from a a high-resolution ocean model study. $B$ is necessary because there is still melt for zero $q_{sg}$. The dependence on $h$ was determined from model experiment with different depths and seems to reflect an acceleration of the melt plume when it rises from greater depths. This is now mentioned in the manuscript.

*p. 4, Fig. 1: This is a very useful and visually attractive figure.*

Thank you!

*p. 5, l. 1: Please give equations showing how TF is computed.*

Done

*p. 5, l. 9: This is a helpful explanation of the interplay between bed topography and thermal forcing.*

Thanks!

*p. 5, Fig. 2: This is another useful figure, which helped me visualize the model forcing, but the caption is not very informative. For instance, the caption for Fig 2a could state that the initial calving front is at x = 0 km and the fjord mouth is at x = 80 km (if I'm interpreting it correctly). For Fig. 2b, please give the source of the data.*

Done

*p. 5, l. 17: "repeat TF of year 2016 until the end of the simulation." I found this confusing. Do you mean that TF from 2016 is the baseline to which the TF anomaly is added?*

This is also something that was mentioned by reviewer #1, we rephrased the sentence as follows: "For 2017 to 2100, as we do not run a coupled model, we repeat the thermal forcing and subglacial discharge of year 2016 until the end of the century, with the anomalies described above."

*p. 6, l. 9: Please see the above comments on model calibration. There is no explanation here of why $\sigma_{max}$ is the specific parameter chosen for calibration, or of why calibration to the observed retreat is preferable to calibration to pre-retreat terminus location.*

Hopefully this is addressed above.

*p. 6, l. 10: Landsat-derived ice front retreat is mentioned here for the first time. I suggest describing the observed retreat, perhaps with a reference to the left column of numbers in Table 1, as part of the background discussion.*

Done

*p. 6, l. 16: Can you say approximately how many CMIP5 models were used to compute this average anomaly, and what is the spread among models?*

We now refer to *Yin et al.* [2011]. They used an ensemble of 19 climate models to quantify this ocean warming in the next two centuries. They found that West Greenland's subsurface ocean temperature would warm by 1.5 degrees on average by 2100, with 5-25-50-75-95th percentiles of 0.5-1-1.5-2.5-4°C. This is now in the text.

*p. 6, l. 17: I'm not sure the 2-degree Paris target is entirely relevant here, given that the target might be exceeded, and we want to know the consequences of missing the target. It would be more appropriate to choose an upper limit based on the CMIP5 spread.*

The Paris agreement is for global air temperature and the ocean is generally slow to respond to an increase in atmospheric temperature. We therefore see a +2°C increase in TF as a high-end

scenario (even though we go up to $+3°$C). We added the spread from *Yin et al.* [2011], keeping in mind that the reported warming is for 2100, and not instantaneous.

*p. 6, l. 24: Can you say why you increase the TF anomaly instantly, instead of phasing it in linearly as might be more realistic? Similarly for SMB/discharge.*

We agree that for projection purposes, it would be more realistic to have a linear increase but we wanted here to do a sensitivity analysis in order to determine the glaciers that are more at risk. In future projection studies based on this work, the forcings will be related to RCP scenarios.

*p. 6, l. 26: It is unclear exactly what is being repeated here.*

Rephrased the sentence.

*p. 7, ll. 3-5: Referring again to the comments above, I'm wondering if a bias correction to the ECCO data would be more defensible than adjusting $\sigma_{max}$. Then you would more likely be capturing the recent retreat for the right reason.*

This is a very good point that was also mentioned by the editor during the initial review. The reason why we do not tune the thermal forcing is that it is a complex time series and we do not know whether we should just optimize for a bias, or if there is a missing trend. Optimizing the threshold was simpler, but we agree that there is no unique solution. We added a sentence in the results section about that.

*p. 8, l. 2: How informative is it that ice front retreat is in good agreement with observations when $\sigma_{max}$ is tuned? Does this suggest that there is something fundamentally "right" about the parameterizations, or does it simply reflect high sensitivity to modest changes in $\sigma_{max}$?*

It is a bit of both, but we would like to highlight the fact that we are comparing here a time series of 2-dimensional ice front positions and, while the terminus position may be on target along a flow line, the model does a surprizingly good job at capturing the pattern of retreat as well. We refer to the study by *Choi et al.* [2018], which compares different calving laws and finds that the one we use here does a reasonable job compared to other existing laws.

*p. 8, l. 9: The text states that the model overestimates the recent retreat of Kakivfaat Sermiat, but Table 1 shows a small underestimate.*

This is a good point. If we look at the figure, we see that on average, the model overestimates the retreat. But for the central flowline that was chosen to make the table, we have a slight underestimation. This is one of the reasons why we have both a table and a figure, as it makes it easier to

compare the retreat qualitatively (figure) and quantitatively (table), but there may be disagreement depending on where the flowline lies. We agree that this may be confusing but did not change the table since this is what is calculated along the flow line.

*p. 10, Fig. 5: This figure nicely illustrates that the TF anomaly is the primary driver of retreat. However, I wondering how the shape of the figure would be different if anomalies were ramped up gradually. Also, I suggest that the caption briefly state what is left out of the fixed-front simulation, such as SMB changes.*

We hopefully addressed the first point above. We added the note in the caption about the SMB that is held constant in all scenarios (including the fixed front one)

*p. 11, l. 17: I couldn't find a description of the control experiment in which the ice front is held fixed.*

We added a description in the Experiment section.

*p. 12, l. 3: The paper makes a strong case for the importance of calving dynamics. At the same time, it does not quantify mass changes due to a more negative SMB, or discuss possible feedbacks of SMB on dynamics. For instance, could a decreasing SMB potentially cause much more mass loss than dynamic retreat? Could SMB-driven thinning significantly modify the calving dynamics (e.g., through reduced tensile stresses)? I realize that SMB changes are beyond the scope of the modeling study, but it would be helpful to talk about them in Section 4.*

This is correct. We keep the SMB constant here, as we want to evaluate the effect of ice/ocean interactions only. The possibility of feedbacks between changes in SMB (primarily surface melt) and calving is indeed interesting to mention. We added a sentence in the discussion about possible feedbacks between SMB and calving.

*p. 12, l. 7: This statement about models with fixed calving fronts seems too general. For example, consider a model without a physically based calving law, in which calving is simply prescribed at the present-day CF. Suppose the model is forced with increasingly negative SMB. This could result in significant thinning, and perhaps ungrounding, of ice all the way to the CF, without necessarily moving the CF. This isn't to say that moving boundaries aren't an improvement, but just to acknowledge that models without moving boundaries may still be able to make useful projections, which might not be overly conservative (especially if SMB dominates the mass balance).*

We agree with the reviewer that models with a fixed ice front will capture some mass loss with an increasingly negative SMB, but these estimates will still be underestimates because of the additional resistive stresses (e.g. basal friction, buttressing, etc). Now, depending on the forcings, the

changes in mass due to SMB may outweigh the changes due to calving front migration. We now limit this statement to the case of ocean warming.

*p. 12, l. 15: I agree that this is an interesting result, which might not have been guessed ahead of time. Given the result, it would be helpful to comment (either here or in Section 2) on the robustness or theoretical justification of alpha and beta.*

We now provide more details as to where the undercutting rate is coming from.

*Technical corrections:*

*p. 2, l. 1: Please be consistent in capitalization of "Northwest" vs. "northwest"*

We now use "Northwest Greenland" and "the northwest coast".

*p. 2, l. 11: don't → do not*

Done

*p. 2, l. 19: Maybe "on the edge" → "on the verge"*

Done

*p. 2, l. 27: To me, "plan-view" suggests 2D in the xy plane. Maybe "3D"?*

It is actually an xy plane, we use a 2d depth-averaged model. We kept "plan-view".

*p. 2, l. 28: "a lot of" → "much". Also p. 3, l. 24.*

Done

*P. 2, l. 33: Define RCP*

Done

*Fig. 1 caption: "are used to calibrate the thermal forcing." Also, capitalize "south" in the figure.*

Done

*p. 5, l. 6: Maybe reword as ". . .the assumption of uniformly distributed melt generates only. . ."*

Done

*p. 5, l. 11: As worded, the subject of "decreases" is "calculation", which isn't intended. Maybe change to "The calculated effective depth"*

Good point! Done

*p. 5, l. 17: "future simulation" → "simulations of future climate"*

Done

*p. 6, l. 7: Hyphenate "real-world"*

Done

*p. 7, Fig. 3 caption: undercutting rate should be m/day instead of m/yr?*

Yes, good catch!

*p. 7,l. 7: 4-¿four*

Done

*p. 9, l. 6: "about 8, 13 and 23 km upstream are distances we find. . ."; awkward wording.*

Rephrased

*p. 9, l. 9: position → positions*

Done

*p. 9, l. 11: "under no warming condition" → "without further ocean warming"*

Done

*p. 10, l. 5: Run-on sentence*

Restructured.

*p. 11,l. 2: "10 km or so"-¿ "∼10 km"*

Done

*p. 11, l. 3: project → projects*

Done

*p. 11, l. 23: advances → advance*

Done

*p. 11, l. 25: retrograde (no hyphen)*

Done

*p. 12, l. 17: sensitive → more sensitive*

Done

[revised manuscript text omitted]

---

## Author Response (AR2)

**Modeling the response of Northwest Greenland to enhanced ocean thermal forcing and subglacial discharge**
**– Response to editor –**

Mathieu MORLIGHEM et al.

February 6, 2019

We thank Andreas Vieli for his comments. We address his remarks below point by point.

*Dear Mathieu Morlighem and co-authors,*

*This manuscript received two in general very positive reviews that highlighted the importance and the advance and extension of the modelling approach to a whole section of the Greenland ice sheet and thereby providing a significant step towards the highly relevant topic of predicting the evolution of ocean-terminating outlet glaciers in Greenland.*
*Both reviewers had however also some critical points and suggestions for improvements, in brief:*
*1) correct all the typos and polish the text (that should have happened before submission), and*
*2) improve and better justify the methodology and description of the methods and experiments.*

*Regarding point 1) all the editing and languages issues as listed by the 2 referees have been corrected and improved, but unfortunately some new typos/text issues have been added with the revised text (see text below). Again, a careful proofreading would have avoided this.*

We indeed found two typos, and we apologize for not catching them earlier. We proofread the text again before submission so hopefully everything should be in order now.

*The authors also added a very useful figure showing the modelled velocity response.*

*Regarding point 2) most points have been addressed satisfactorily and carefully or justified why nothing has been changed. Regarding some of the valuable questions of referee 2 on the modelling approach/experiment choice (see details in list below) have also been well explained and justified in the response, but as these are general points that are relevant for all readers, I recommend to*

*add a few brief sentences of the justification (see details in list below) also in the text of the paper (not just in the author-response). Specifically, these points concern:*

*2a) comment by referee 2 regarding why 'calibration of $\sigma_{max}$ for retreat rather than pre-retreat positions': a satisfying explanation is given in response but not added to the text of the paper. This is relevant for all readers not just the referee so please clarify this also in the paper-text (maybe in section 2.3 calving parametrization; something along the lines of your explanation in the response of: 'Another reason is that stable glaciers generally have their terminus on a distinct feature (ledge, ridge, etc) in the bed topography and the numerical model is also stable for a wide range of $\sigma_{max}$. Constraining the calving threshold, $\sigma_{max}$, is easier to do for retreating calving front as we constrain the rate of retreat as opposed to just the stability of the glacier. This is something that we noticed in Morlighem et al. [2016] and Choi et al. [2018].').*

We indeed debated whether this should be in the main text or left only in the response, but we agree with the editor that it may be of interest to some readers. We added the following: "Another possible approach would be to calibrate $\sigma_{\mathrm{max}}$ during a period of ice front stability. One of the problems with this alternative approach is that stable glaciers generally have their termini on distinct basal features, such as ridges or ledges. The numerical model is also stable for a wide range of $\sigma_{max}$ under these conditions, as shown in *Morlighem et al.* [2016] and *Choi et al.* [2018]. The threshold $\sigma_{max}$ is easier to calibrate for retreating glaciers, as it directly constrains the rate of retreat."

*2b) comment by referee 2 regarding stepwise increase of thermal forcing anomaly and how different would results for linear/gradual forcing be. I can accept this choice of step changes and I am happy with the justification in the response, but please add this explanation/justification (sensitivity analysis rather than projection, to assess glaciers at risk and bulk magnitude of mass loss) also in the paper-text (in section '2.4 Experiments').*

We added "While a gradual increase in ocean thermal forcing and subglacial discharge would be more realistic, we want here to perform a sensitivity analysis in order to determine the glaciers that are more at risk." to the Experiments section, as suggested.

*Thus overall this manuscript is very close to publication but these 2 points above regarding methods/experiment choice should be addressed in addition to the general technical/minor points listed in detail below. And please very carefully proofread the final manuscript again! I congratulate the authors for their very interesting paper and also would like to thank them for their supportive handling of the revision process.*

Thank you, and thank you for the very careful editing of this paper!

*Rather minor technical and editing issues*

*p. 3 line 25: maybe 'The model mesh comprises...' is more elegant.*

Done

*p. 5 line 12: a 'the' is missing here: '...is the temperature of the local freezing point...'*

Done

*p. 5 line 15: delete one of the 'a' before 'high-resolution'.*

Done

*p. 5 line 16: style: maybe 'ocean modelling study' is better.*

Done

*p. 5. Line 17: plural 'experiments'*

Done

*p. 7. Fig. 3 caption: '...(m/day) and calibrated...'*

Done

*p. 8 line 9-10: there is a 'for' missing: '... as for all other experiments'.*

*p. 8 line 13-14: awkward formulation, maybe simplify to: '...an artefact that is due to our underestimation of the rate of undercutting for this region'.*

Done

*p. 8. Line 25: something wrong in this sentence, maybe change to: '...the ice front positions, that were manually digitized from Level 1 Landsat imagery, together with the modeled...'*

Done

*p. 9 table 1 in caption: add that positions are along the flowline/centerline: '...observed and modelled ice front retreat (in km along the flowline9...'*

Done

*p.10 fig 4 caption: the 4th word (modelled) should be 'lower case'*

Done

*p. 10 line 10: '...upstream of the 2007 position',...'*

Done

*p. 11 Fig 5 caption: should be in plural: 'Modeled ice velocities (solid lines) and ice front positions (dashed vertical lines)...'*

Done

*p. 13 line 3: style '...mass loss is significantly higher...'*

Done

*p. line 13: either consistent plural or single tense: '...any glaciers which advance.' or 'any glacier which advances'.*

Done

*p. 13 line 29: simplify to '...consistently under-estimates ice sheet mass loss...'*

Done

*p. 13 line 30: '...should therefore be treated with ...'*

Done

*p. 14 line 5: should it not be '...towards ice-ocean-climate coupled model ...'*

Done

*p. 14 line 9: I think there is a 'more' missing: '...is itself more sensitive to...'*

Done

*p. 14 line 14: '...no numerical ocean model...'*

Done

[revised manuscript text omitted]